# Extreme intratumour heterogeneity and driver evolution in mismatch repair deficient gastro-oesophageal cancer

Katharina von Loga [1,6,8], Andrew Woolston [1,8], Marco Punta[2], Louise J. Barber[1], Beatrice Griffiths[1], Maria Semiannikova[1], Georgia Spain[1], Benjamin Challoner [1], Kerry Fenwick[3], Ronald Simon[4], Andreas Marx[4,7], Guido Sauter[4], Stefano Lise [2], Nik Matthews[3] & Marco Gerlinger [1,5]*

Mismatch repair deficient (dMMR) gastro-oesophageal adenocarcinomas (GOAs) show better outcomes than their MMR-proficient counterparts and high immunotherapy sensitivity. The hypermutator-phenotype of dMMR tumours theoretically enables high evolvability but their evolution has not been investigated. Here we apply multi-region exome sequencing (MSeq) to four treatment-naive dMMR GOAs. This reveals extreme intratumour heterogeneity (ITH), exceeding ITH in other cancer types >20-fold, but also long phylogenetic trunks which may explain the exquisite immunotherapy sensitivity of dMMR tumours. Subclonal driver mutations are common and parallel evolution occurs in *RAS*, *PIK3CA*, SWI/SNF-complex genes and in immune evasion regulators. MSeq data and evolution analysis of single region-data from 64 MSI GOAs show that chromosome 8 gains are early genetic events and that the hypermutator-phenotype remains active during progression. MSeq may be necessary for biomarker development in these heterogeneous cancers. Comparison with other MSeq-analysed tumour types reveals mutation rates and their timing to determine phylogenetic tree morphologies.

[1] Translational Oncogenomics Laboratory, Centre for Evolution and Cancer, The Institute of Cancer Research, London SW3 6JB, United Kingdom. [2] Bioinformatics Core, Centre for Evolution and Cancer, The Institute of Cancer Research, London SM2 5NG, United Kingdom. [3] Tumour Profiling Unit, The Institute of Cancer Research, London SW3 6JB, United Kingdom. [4] Institute of Pathology, University Medical Center Hamburg-Eppendorf, 20246 Hamburg, Germany. [5] Gastrointestinal Cancer Unit, The Royal Marsden Hospital, London SW3 6JJ, United Kingdom. [6] Present address: Biomedical Research Centre, The Royal Marsden Hospital, London SM2 5PT, United Kingdom. [7] Present address: Institute of Pathology, University Hospital Fuerth, 90766 Fuerth, Germany. [8] These authors contributed equally: Katharina von Loga, Andrew Woolston. *email: marco.gerlinger@icr.ac.uk

Gastro-oesophageal adenocarcinomas (GOAs) are one of the commonest causes of cancer mortality worldwide[1]. Microsatellite instable (MSI) and DNA mismatch repair deficient (dMMR) cancers are a distinct subtype of GOAs with a prevalence of up to ~20% in the stomach and gastro-oesophageal junction[2–4]. dMMR results from genetic inactivation of *MLH1*, *MSH2, MSH6, PMS2* or methylation of *MLH1*. These tumours are characterized by a hypermutator-phenotype leading to high mutation loads and a large fraction of small insertions and deletions (indels), predominantly in homopolymer and dinucleotide repeats. dMMR GOAs have distinct clinical characteristics compared to their MMR-proficient counterparts, including lower stage in the UICC TNM classification of malignant tumours at presentation and better survival[3]. This has been attributed to a large number of mutation-encoded neoantigens, which enable recognition by the adaptive immune system. Consistent with the notion of high immunogenicity, dMMR cancers are among the tumour types most sensitive to checkpoint-inhibiting immunotherapy (85.7% response rate in small series)[5,6]. However, not all tumours respond to immunotherapy and some acquire resistance after initial benefit. Chemotherapy and anti-angiogenic drugs are the only other systemic treatment options for dMMR GOAs and the identification of novel therapeutics is important to improve outcomes.

Genetic intratumour heterogeneity (ITH) and ongoing cancer evolution have been demonstrated in multiple cancer types[7]. The ability to evolve is thought to foster cancer progression, drug resistance and poor outcomes[8]. High mutation rates may fuel evolvability by generating an abundance of novel phenotypes which selection can act upon[9]. A pan-cancer study indeed demonstrated large numbers of subclonal mutations within single tumour regions of MSI cancers[10]. However, it has not been investigated in dMMR GOAs whether the MSI hypermutator-phenotype remains active during progression, how this impacts ITH and phylogenetic trees, and whether subclonal driver mutations evolve. Our previous work in kidney cancer for example showed that most driver mutations are located in subclones[11]. Subclonal driver mutations are poor therapeutic targets as co-existing wild-type subclones remain untargeted[12]. They furthermore hinder effective biomarker development as the analysis of single tumour regions incompletely profiles the genomic landscape of the entire tumour. Large-scale sequencing analyses of MSI GOAs identified *TP53*, *RNF43*, *ARID1A, PIK3CA, KRAS* and *PTEN*, as the most frequently altered driver genes[13]. Mutations in antigen presentation (*MHC, B2M*)[2] and interferon signalling pathway (*JAK1/2*)[14,15] genes also frequently occur in MSI tumours and they have been suggested to enable immune evasion[2]. However whether they are truncal or subclonal within individual tumours is unknown.

Multi-region exome sequencing (MSeq) reconstructs cancer evolution by comparing mutational profiles from spatially separated tumour regions. MSeq found that mutations often appear to be present in all cancer cells (i.e. clonal) in a single tumour region even if they are absent from other regions of the same tumour[11,16]. Spatial constraints in solid tumours that preclude intermixing of evolving subclones likely explains this 'illusion of clonality' phenomenon when heterogeneity is only investigated in a single sample per tumour[17,18]. We apply MSeq to four surgically resected GOAs showing dMMR on immunohistochemistry and combine this with subclonality analysis of single tumour biopsies from 64 MSI GOAs sequenced by The Cancer Genome Atlas (TCGA)[2] to assess ITH and the evolution of these tumours.

## Results

**Samples**. Seven primary tumour regions from each of four GOAs (Fig. 1a) were subjected to MSeq with a target depth >200×

(Supplementary Data 1). Two lymph node metastases were included from each of two cases. TNM-stage was assessed but no other clinical information was available as the samples had to be anonymised to comply with local ethics and research legislation. Absence of MLH1 and PMS2 staining and positive staining for MSH2 and MSH6 (Fig. 1b), indicated *MLH1* deficiency. No known Lynch syndrome mutations in *MLH1*, *MSH2/6* or *PMS2* were identified in DNA from non-malignant tissue, confirming that these were sporadic dMMR tumours.

**Mutational intratumour heterogeneity**. About 1518–4148 (median: 1814) non-silent mutations were identified per case (Fig. 1c). The high mutation burden and the large fraction of indels (20–34%) were consistent with an MSI-phenotype[2]. The number of ubiquitous non-silent mutations that were detected across all sequenced regions per tumour ranged from 329 to 1006 (median: 702). This exceeded the number of ubiquitous non-silent mutations reported for clear cell renal cell carcinomas (ccRCC, median: 28)[11] and even for lung cancers (median: 137)[16] and melanomas (median: 436)[19], which are among the most highly mutated cancer types[20] (Fig. 1d). The difference was significant between dMMR GOA and lung and ccRCC but not for melanomas. MSeq-identified ubiquitous mutations are likely to define the mutations that were present in the founding cell of each tumour before diversification into subclones occurred[11]. These high numbers hence reveal that the dMMR-phenotype was likely acquired in the precancerous cell lineage considerably earlier than malignant transformation of the founding cell. Malignant transformation shortly after dMMR acquisition which was then followed by selective sweeps is an alternative explanation. Yet, it appears unlikely that this would have left no trace of the early subclones in any tumour.

A median of 1194 mutations were only detectable in some but not in all analysed tumour regions per case and hence heterogeneous. This significantly exceeded the heterogeneous mutation burden detected by MSeq in ccRCC[11] by 24-fold, in lung cancer[16] by 40-fold, and in melanoma[19] by 32-fold (Fig. 1d). Importantly, the median mutation load per region in these MSeq series was similar to those reported by the TCGA for the respective cancer type (Fig. 1e), suggesting that these small series are reasonably representative of each tumour type. Thus, dMMR tumours are characterized by extreme ITH compared to other cancer types.

High mutation and neoantigen loads are associated with immunotherapy benefit. Recent data suggested more specifically that a high burden of clonal mutations/neoantigens is important for immunotherapy success[21,22]. Applying the NetMHC algorithm predicted 1120–3052 strong class I MHC binding neoantigens per tumour (Supplementary Fig. 1). Between 215 and 926 of these were clonal. This is higher than clonal neoantigen loads reported for most lung cancers or melanomas[23]. It is conceivable that this high clonal neoantigen burden explains the immunotherapy sensitivity of dMMR tumours[21].

**Mutational signatures reveal processes driving evolution**. We next investigated mutational signatures by counting the number of all possible base substitution in their trinucleotide contexts (Supplementary Fig. 2) and assigning these to 30 mutational signatures[20] (Fig. 1f). The COSMIC mutational signatures 6 and 15 are characteristic for MSI cancers and these were abundant among ubiquitous and heterogeneous mutations. Signature 1 mutations reflect the spontaneous deamination of methylated cytosine, a mutational process active in most normal tissues. Signature 1 was detected in 17–52% (219–449 mutations in absolute number) of ubiquitous mutations. A fraction of these

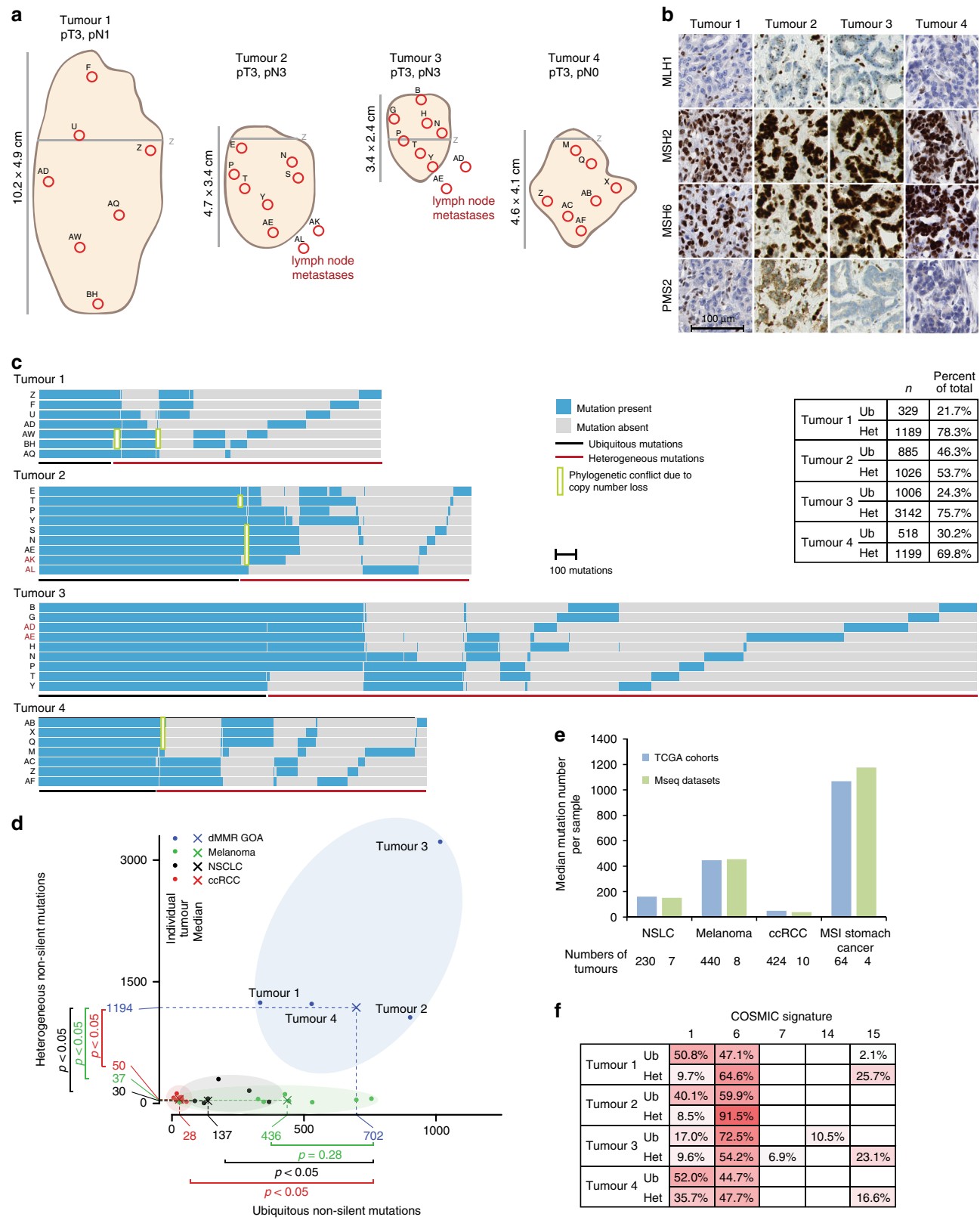

were likely acquired in the normal cells over the lifetime of these patients. However, based on the estimated mutation rate in normal gastro-oesophageal epithelium, only 0.5–1 signature 1 mutations would be expected to accumulate per lifeyear[24–26]. It is hence likely that the dMMR-phenotype also contributes to the generation of signature 1 mutations. This is further supported by

9–10% of the subclonal mutations in Tumours 1–3 and 36% in Tumour 4 showing signature 1 and consistent with a recently suggested role of the MMR-system in the repair of deamination defects[27]. A total of 10.5% of the ubiquitous mutations in Tumour 3 showed signature 14, which has been described in dMMR cancers that are also *POLE* or *POLD1* mutant[28]. Tumour 3

**Fig. 1 Intratumour heterogeneity of somatic mutations. a** Tumour size, location, TNM-stage and regions selected for sequencing. The grey line labelled (Z) marks the gastro-oesophageal junction. **b** Immunohistochemical staining of MLH1, MSH2, MSH6 and PMS2. **c** Heat maps showing the presence (blue) or absence (grey) of non-silent somatic mutations that were identified by MSeq across tumour regions. The table shows the number of heterogeneous (Het) and ubiquitous (Ub) mutations identified in each tumour and their percentage of the total non-silent mutation count of the tumour. **d** Comparison of ubiquitous and heterogeneous mutation counts across four different tumour types analysed by MSeq (dMMR GOA: mismatch repair deficient gastro-oesophageal adenocarcinoma, Melanoma, NSCLC: non-small cell lung cancer, ccRCC: clear cell renal cell carcinoma). The Mann–Whitney test was used to assess significant differences in mutation loads between dMMR GOA and other tumour types. **e** Median mutation loads of individual regions from MSeq datasets compared to the median single sample mutation loads from the Cancer Genome Atlas KIRC, SKCM, STAD and LUAD cohorts. **f** COSMIC mutational signature analysis of ubiquitous (Ub) and heterogeneous (Het) mutations in four dMMR GOAs. Non-silent and synonymous mutations were included in the analysis and only signatures which contributed to ≥5% of mutations in at least one sample are shown.

harboured a *POLD1* mutation but this was subclonal and could not explain the presence of clonal signature 14 mutations. The absence of signature 14 from subclonal mutations furthermore suggested that this is a passenger mutation. No other mutational signatures contributed substantially to the heterogeneous mutations, confirming that the MSI-phenotype remains active during cancer progression and is the primary mechanism generating these large numbers of subclonal mutations.

**The evolution of copy number aberrations**. DNA copy number aberration (CNA) profiles revealed near-diploid profiles across all regions of Tumours 2 and 3 (Fig. 2a and Supplementary Fig. 3). Tumour 4 showed highly aberrant near-tetraploid profiles in all regions. A high number of mutations were present on all copies of the major allele of most gained chromosomes (Fig. 2b), indicating that whole genome duplication and chromosomal instability (CIN) had occurred late on the trunk of the phylogenetic tree in Tumour 4. CIN was confirmed by the weighted genome integrity index (wGII) that measures the proportion of all chromosomes with copy number states that differ from the ploidy of a sample and where values above 0.2 support the presence of CIN[29] (Fig. 2a). Near-diploid and near-triploid CNA profiles were found in distinct regions of Tumour 1. Together with an increase in wGII from ~0.2 in the near-diploid regions to >0.5 in near-triploid regions and the occurrence of new CNAs in individual tumour regions, this revealed the acquisition of subclonal CIN during cancer progression. All four lymph node metastases were near-diploid with wGII values ≤0.2, demonstrating that CIN, which has been associated with tumour aggressiveness in several cancer types including GOA[28], is not required for metastasis formation.

We next investigated which specific CNAs were ubiquitous/clonal and had hence occurred early in the evolution of these dMMR tumours (Fig. 2c and Supplementary Fig. 3). Ubiquitous Chr17p, Chr18 and Chr22 loss of heterozygosity (LOH) were each present in two tumours. Ubiquitous LOH of Chr3p, Chr5q and Chr17p encompassed tumour suppressor genes, which are recurrently mutated in dMMR GOAs[2] (*MLH1*, *APC* and *TP53*). Among the small number of ubiquitous gains, only Chr8q and Chr20q were gained in more than one tumour. To further time the acquisition of these recurrent truncal CNAs, we mapped ubiquitous mutations onto the allele-specific CNA profiles. Copy number gains that occurred early can be identified if the majority of mutations in that region have a mutation copy number[23] which is lower than that of the gained allele. The Chr8 gain in Tumour 2 and the Chr8q gain in Tumour 4 (Fig. 2d), but not Chr20 gains (Fig. 2e), showed a near complete absence of mutations on all copies of the gained allele and were hence acquired on the phylogenetic trunk before or soon after the MSI-phenotype. Thus, Chr8q gains, which are the commonest CNAs in MSI GOAs[2], can be among the earliest genetic aberrations in these tumours.

**Reconstruction of tumour phylogenies**. We next deconvoluted the subclonal composition of individual regions and reconstructed the phylogenetic tree for each tumour (Fig. 3). Similar to MSeq analyses of other tumour types[11,16,19], this revealed branched evolution. Comparison of the phylogenetic trees with the mutation heatmaps showed some phylogenetic conflicts. Inspection of the CNA status of the mutated DNA positions showed that most conflicts could be explained by losses of chromosome copies in individual regions (marked in green in Fig. 1c and Supplementary Fig. 4). Thus, subclones can lose a small proportion of mutations during cancer evolution.

Phylogenetically closely related clones were usually located in close physical proximity (Supplementary Fig. 5), indicating that cell motility is limited and that these tumours evolve in a spatially ordered fashion. Importantly, each of the two lymph node metastases analysed in Tumours 2 and 3 had evolved from distinct subclones rather than being seeded by the same subclone or sequentially from one node to the other (Fig. 3). Dissemination hence propagated subclonal diversity from the primary tumour to metastatic sites. In addition, subclonal mutations, defined as private mutations estimated to be present in ≤70% of the cancer cells of a sample, were detectable within three metastatic sites with good cancer cell content (Supplementary Table 1). Subclonal mutations within lymph nodes were again predominated by the MSI-specific mutational signatures 6 and 15 (Supplementary Table 2). Thus, the dMMR-phenotype continues to generate ITH in metastases.

**Identification of truncal drivers**. We next assessed the evolution of putative driver mutations and of corresponding LOH of tumour suppressor genes and mapped them onto the phylogenetic trees (Fig. 3 and Supplementary Data 2). A frameshift mutation and LOH of *MLH1* occurred on the trunk of Tumour 1, consistent with biallelic *MLH1* loss. No genetic aberrations of *MLH1* were detectable in Tumours 2–4 but qPCR confirmed hypermethylation of the *MLH1* promoter as the cause for dMMR in these cases (Supplementary Fig. 6)[30]. Tumours 2–4 furthermore harboured a truncal frameshift mutation in *MSH6*. Mutations in the histone methyltransferase and tumour suppressor gene *PRDM2*, one in combination with LOH of the second allele were also truncal in all four cases and truncal frameshift mutations of the TGFβ signalling regulator *ACVR2A* were detected in three cancers. Both genes have been suggested as likely drivers in MSI GOAs[13].

One tumour showed a disrupting mutation and LOH of *ARID1B* and two tumours each harboured two truncal mutations in *ARID1A*, which are all members of the SWI/SNF-chromatin-modifying complex. We could not formally demonstrate that the two mutations affected both alleles of the *ARID1A* tumour suppressor gene but biallelic inactivation is likely as all mutations were disrupting in nature, suggesting evolutionary selection for inactivating events. A frameshift mutation and LOH of *PBRM1*, a

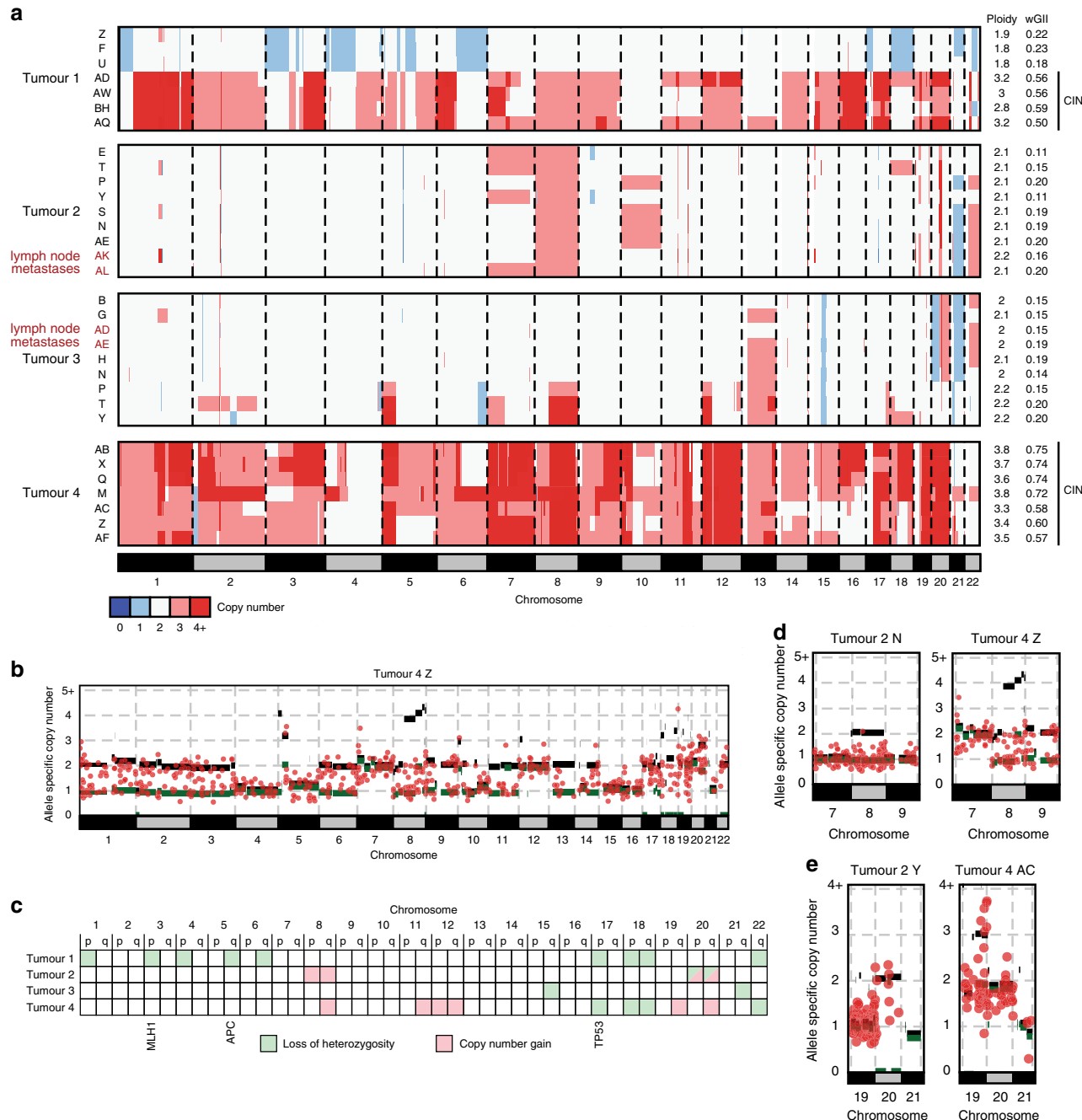

**Fig. 2 Intratumour heterogeneity of DNA copy number aberration. a** Genome-wide DNA copy number profiles of each tumour region. Profiles showing chromosomal instability (CIN) are labelled with a black bar on the right. **b** Example of an allele specific DNA copy number profile and superimposed copy numbers of somatic mutations from Tumour 4. This allows timing of CIN/genome duplication, demonstrating late acquisition, as large numbers of mutations are located on the major alleles for most gained chromosomes. **c** Ubiquitous loss of heterozygosity (LOH) or copy number gains identified in each of the four tumours. Tumour suppressor genes commonly mutated in dMMR GOAs and which are located on chromosomes showing ubiquitous LOH are labelled. **d** Examples of the allele specific copy number and copy number of corresponding somatic mutations for Chr8 and **e** for Chr20 which showed recurrent ubiquitous gains in our series.

further SWI/SNF-complex member, co-occurred with biallelic *ARID1B* loss on the trunk of Tumour 1. This emphasizes an important role for SWI/SNF-complex aberrations in dMMR GOA development.

Truncal mutations in *TP53* were found in three tumours. Tumours 1 and 4 also showed LOH, leading to biallelic *TP53* inactivation. These specific cancers had undergone genome duplication and acquired CIN, consistent with a permissive role

of *TP53* loss for CIN[31]. Moreover, both showed truncal Chr18q loss which promotes CIN in colorectal cancer[32]. *TP53* inactivation and Chr18q loss may hence predispose tumours to subsequently evolve CIN. Frameshift mutations of *RNF43*, a negative regulator of the APC/β-catenin-pathway that frequently acquires heterozygous mutations in MSI tumours[33], were present in three tumours. The tumour without an *RNF43* mutation harboured two truncal mutations in the *APC* tumour suppressor

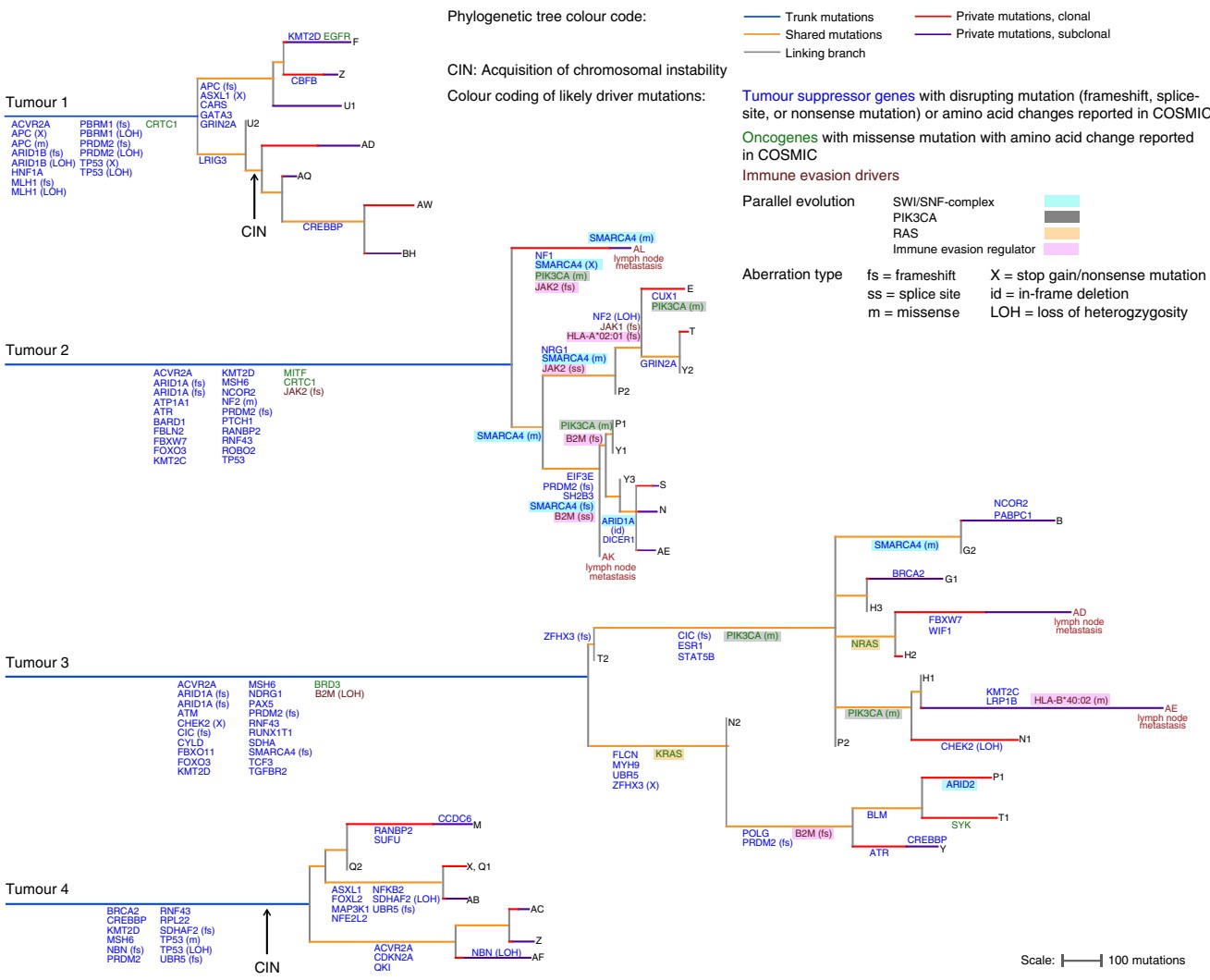

**Fig. 3 Tumour phylogenetic trees.** Trees were reconstructed from non-silent and synonymous mutations and trunk and branch lengths are proportional to the number of mutations acquired. Trees are rooted at the germline DNA sequence, determined by exome sequencing of DNA from tumour adjacent normal tissue. Subclones that define the tips of the tree are labelled with the tumour region in which they were identified. Numbers were added where several subclones were identified by the phylogenetic deconvolution algorithm within a tumour region, with 1 defining the largest intra-regional subclone and 2, 3 increasingly smaller subclones. Private mutations were furthermore split into those that were clonal in the analysed region (present in >0.7 of the cancer cell fraction in that region) and those that were subclonal (present in ≤ 0.7 of the cancer cell fraction). Likely driver mutations and relevant loss of heterozygosity (LOH) events were mapped onto the branch of the trees where they likely occurred. Genes affected by more than one genetic aberration within a tumour are labelled with the genetic aberration type that occurred. Arrows labelled 'CIN' indicate the likely onset of chromosomal instability.

gene as an alternative mechanism of β-catenin activation. Together, aberrations in *TP53*, the SWI/SNF-complex, *PRDM2*, dMMR-, APC/β-catenin signalling- and TGFβ signalling-genes each occurred on the phylogenetic trunks of at least two cases.

**Parallel evolution.** Assessing heterogeneous driver mutations revealed striking examples of parallel evolution, a strong signal that these evolved through Darwinian selection:[7,17,34,35] Tumour 2 acquired five subclonal mutations in *SMARCA4*, encoding a catalytic subunit of the SWI/SNF-complex. These had occurred in addition to two truncal mutations (M274fs, K1071fs) in *ARID1A*. A third *ARID1A* mutation was subclonal and affected recurrently mutated amino acids (AA163-164del) located proximally to the truncal frameshift mutations. This may be functionally relevant if *ARID1A* had retained some residual activity despite the more distal mutations. Parallel evolution of five subclonal *SMARCA4* mutations in this tumour with truncal *ARID1A* mutations suggests that SWI/SNF-complex aberrations are not only important

for carcinogenesis but that progressive inactivation may contribute to cancer progression.

A *PIK3CA* hotspot mutation (H1047R) was detected in P1 and Y1 but also in the distantly related subclone AL in Tumour 2. Copy number changes that could explain a loss of this mutation in subclones with wild-type *PIK3CA* were absent (Supplementary Fig. 3). The most parsimonious explanation for this phylogenetic conflict is that the same mutation independently evolved twice, once in AL and once in the ancestor cell of P1 and Y1. Intuitively this may appear unlikely, but a tumour of this diameter contains >10 × 10$^9$ cancer cells[9] that have undergone approximately the same number of cell divisions to grow to this size from the founding cell. It is conceivable that two cells independently acquire the same mutation in some tumours of this size. With one further *PIK3CA* hotspot mutation in region E (Y1021C), this identified three *PIK3CA* parallel evolution events in Tumour 2.

Mutations in the SWI/SNF-complex members *SMARCA4* and *ARID1A* were present on the trunk of Tumour 3. Additional

SWI/SNF mutations, one in *ARID2* and one in *SMARCA4*, evolved in subclones, the latter potentially complementing monoallelic *SMARCA4* loss on the trunk to biallelic inactivation in the subclone. Further parallel evolution was apparent in Tumour 3 based on the acquisition of *KRAS* (G13D) and *NRAS* (G12C) oncogenic mutations in distinct subclones. Two hotspot *PIK3CA* mutations (E418K, Y1021H) sequentially occurred in one clade of Tumour 3.

The tumour suppressor gene *PRDM2* harboured frameshift mutations on the trunks of Tumours 2 and 3 and a second frameshift mutation was acquired in subclones of each tumour, potentially leading to biallelic inactivation. Subclonal inactivating mutations of the cell cycle regulator and DNA damage repair genes *CHEK2*, *ATR* and *BLM* occurred in Tumour 3. Together with truncal LOH of *CHEK2*, both alleles of this gene were inactivated. Heterozygous *BLM* and *ATR* mutations may be functionally relevant as both genes show haploinsufficiency[36,37].

Given the high burden of mutations caused by dMMR, it is possible that several mutations which we classified as likely drivers are passengers without significant fitness effects. However, parallel evolution and the strong functional evidence for driver status of the identified *KRAS*, *NRAS* and *PIK3CA* mutations and of inactivating mutations in SWI/SNF-complex members in cancer[38] support the functional relevance of these specific aberrations.

**The evolution of immune evasion drivers**. Tumour 2 harboured a truncal *JAK2* frameshift mutation. In addition, a subclonal *JAK2* splice-site mutation evolved in one clade and a frameshift mutation in region AE. Another subclone had acquired a *JAK1* frameshift mutation but no evidence for biallelic inactivation was found. A subclonal frameshift mutation was present in *HLA-A*02:01* (Supplementary Data 3). Assessing the neoantigens binding to this HLA allotype revealed that this could lead to a 12% reduction in the number of neoantigens presented by these subclones (Supplementary Fig. 7). One clade in Tumour 2 furthermore acquired two disrupting mutations in *B2M*. Inspecting short read sequencing data confirmed that these were not located on the same allele but conferred biallelic inactivation which abrogates MHC Class I antigen presentation (Supplementary Fig. 8).

LOH of *B2M* was present on the trunk in Tumour 3 and a *B2M* frameshift mutation was acquired in a subclone, also establishing biallelic *B2M* loss. Although several primary tumour regions in Tumours 2 and 3 showed biallelic *B2M* inactivation this was not propagated to any of the four lymph node metastases (Fig. 3). The lymph node metastasis AE in Tumour 3 acquired a missense mutation in *HLA-B*40:02* (Supplementary Data 3) with unknown functional impact. If this *HLA-B*40:02* mutation compromised antigen presentation, 12% of neoantigens could no longer be presented. In contrast to lung cancers which are frequently chromosomally unstable and acquire subclonal LOH of *HLA* genes as immune evasion mechanisms[39], no such LOH events were identified (Supplementary Data 3).

To investigate why immune evasion drivers only evolved in 2/4 tumours, we assessed cytotoxic CD8+ T-cell infiltrates by immunostaining. The two tumours with evidence of immune evasion events, which also had the highest truncal and subclonal mutation burdens, showed higher T-cell infiltrates than the other two cases (Fig. 4). dMMR GOAs with high immunogenicity and T-cell infiltrates may hence be particularly prone to subclonal immunoediting.

**Darwinian selection over time**. The ratio of non-synonymous mutations to synonymous mutations (dN/dS-ratio) has been used to estimate positive and negative selection in cancer[40]. dMMR tumours have high clonal but also subclonal mutation burdens and we reasoned that this may enable applying these ratios to evaluate how selection changes from truncal mutations to subclones. dN/dS ratios were close to 1 for the truncal mutations of all cases (0.95–1.06), indicating that the majority of mutations are neither under positive nor under negative selection. However, the dN/dS ratios increased to 1.16 in Tumour 1 and 1.31 in Tumour 2 for private mutations, indicating positive selection (Fig. 5 and Supplementary Table 3). Together with the identification of parallel evolution in Tumours 2 and 3, this suggests that these tumours are under selection pressure and adaptive mutations continue to evolve. The dN/dS <1 in the shared mutations of Tumour 4 may be a sign of negative selection during early evolution. Our results show that MSeq allows to dissect the temporal dynamics of selection in dMMR tumours and this can be used to reveal what genetic alterations are selected for or against in larger series.

**Multi-region vs. single-region heterogeneity analysis**. Our next aim was to gain further insights into the evolution of dMMR GOAs by deconvolution of clonal and subclonal mutations in single samples from the TCGA GOA dataset[2].

We first used our MSeq dataset to assess which information can be robustly generated by single sample deconvolution and which ones are more likely to be gained by MSeq. The total mutation load in a single sample exceeded the MSeq-determined ubiquitous/truncal mutation load by an average of 73% across the four tumours (Fig. 6a). Following bioinformatic deconvolution of regional mutations into clonal and subclonal, the average clonal mutation burden determined in single samples still exceeded the number of mutations identified as ubiquitous by MSeq by 34%. Moreover, the number of mutations identified as clonal in a single region varied highly between samples from the same tumour. This could not be attributed to different cancer cell contents as no correlation was observed (Supplementary Fig. 9).

We furthermore assessed whether the parallel evolution mutations, that have a high probability of being actual drivers and were found to be subclonal by MSeq analysis, could also have been accurately identified as subclonal by single-region analysis. Only 40% of *B2M* mutations that were subclonal based on MSeq were accurately identified as subclonal in individual regions whereas 60% appeared clonal (Fig. 6b, c). This illusion of clonality in single sample analysis also affected 40% of *JAK2* mutations, 76.2% of *SMARCA4* mutations, 66.7% of *RAS* mutations and 35.7% of *PIK3CA* mutations. Overall, 59.0% of these likely driver mutations appeared clonal in single-region analysis despite clear subclonal status based on MSeq. This supports the conclusion from MSeq studies in other tumour types, that single-region analysis overestimates the clonal dominance of driver mutations[11,16].

We next analysed 64 MSI GOAs cancers from TCGA. All samples harboured subclonal mutations but only a median of 21.3% of mutations were subclonal (Fig. 6d) compared to a median of 60.1% in MSeq data. We then assessed the clonality of mutations in driver genes which we had found to be either predominantly clonal or subclonal by MSeq. The highest frequency of subclonal mutations was found in *ARID2* and *SMARCA4* whereas *ACVR2A* was almost always clonal in TCGA data (Fig. 6e), consistent with MSeq data where these occurred late and early, respectively. Mutations in the remaining driver genes were predominantly clonal in TCGA data, but in light of our MSeq data this is likely limited by the overestimation of clonal status in single-region analysis.

Only 2/64 TCGA cases showed parallel evolution of two subclonal *SMARC4* mutations, each, and two subclonal *PIK3CA*

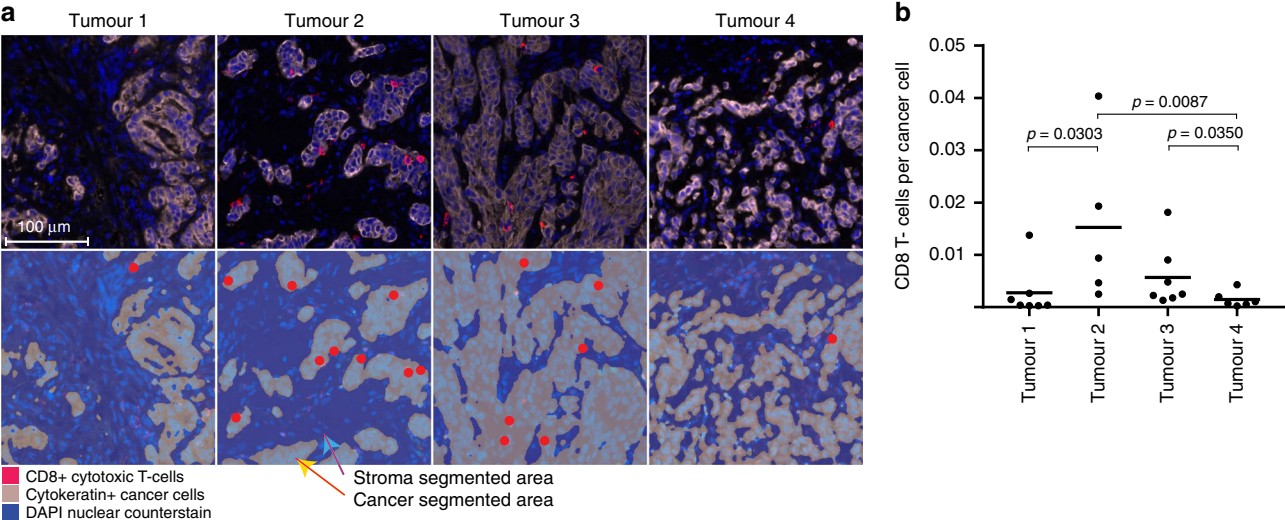

**Fig. 4 CD8+ T-cell infiltrates in dMMR GOAs. a** Representative images of CD8+ T-cell infiltrates in Tumours 1–4. Upper row: fluorescent composite IHC image showing cancer cells and CD8+ cytotoxic T-cells. Bottom row: segmentation of cancer areas (grey, yellow arrow) and stroma (blue, light blue arrow) allowed to only count the highlighted CD8+ T-cell in cancer areas. **b** The ratio of CD8+ T-cells divided by the number of cancer cells (cytokeratin-positive cells) for all regions of Tumours 1–4. Black bar: median; *p*-values (Spearman rank test) are shown for significant differences.

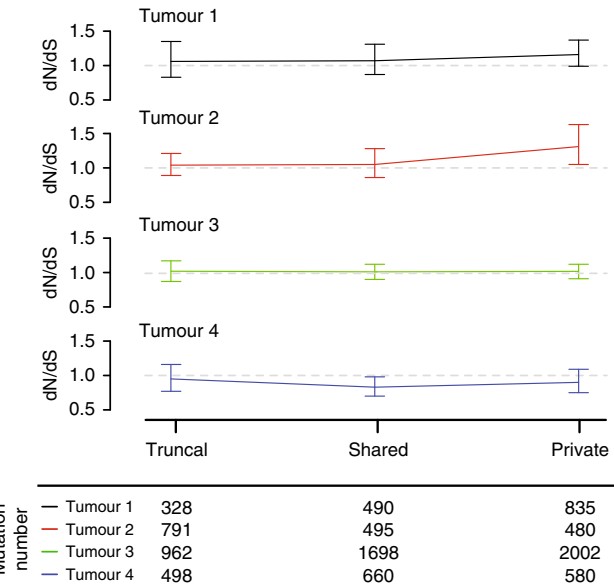

**Fig. 5 Non-synonymous to synonymous (dN/dS) mutation ratios.** dN/dS ratios for ubiquitous, shared and private mutations, adjusted for common mutation biases. Error bars show 95% CI's. The total number of synonymous and non-synonymous mutations available in each category for the analysis are shown beneath the plot.

mutations evolved in one case. No parallel evolution of driver mutations in *RAS* or immune evasion regulators was identified. Together with the detection of parallel evolution in spatially distinct tumour regions by MSeq, this illustrates the limitation to identify such events by single sample analysis. Two independent disrupting mutations in *ARID1A* were found to be clonal in each of 16/64 tumours (25%) and only four tumours had one clonal and one subclonal inactivating event. This confirms frequent biallelic inactivation.

Clonal and subclonal mutations in TCGA samples were dominated by the MSI-specific mutational signatures 6 and 15

(Fig. 6f, g), confirming our MSeq results. A total of 44.0% of clonal mutations displayed signature 1 and although this significantly decreased among subclonal mutations, it remained the second most abundant mutation signature. Together with a significant increase in signature 15 among subclonal mutations, this supports the change in mutational processes between early progression and subclonal diversification as seen in the MSeq dataset. Timing of copy number changes in the TCGA dataset supported that chromosome 8 gains had been acquired before or early after the MSI-phenotype in ~60% of cases (Fig. 6h and Supplementary Fig. 10).

**Mutational mechanism and their timing influence phylogenies.** To investigate how mutational processes and their timing influence phylogenetic tree morphologies, we represented dMMR GOAs, melanomas[19], lung[16] and renal cancers[11] as a single phylogenetic tree with a branching structure similar to those revealed by MSeq and by using the average number of ubiquitous and heterogeneous mutations (Fig. 1d) to scale trunk and branch sizes (Fig. 7). This revealed that dMMR leads to long trunks even exceeding the trunk size of carcinogen-induced cancers (UV light in melanomas, cigarette smoke in lung cancer). Additionally, dMMR tumours showed prominent branches, whilst branch lengths in lung cancer and melanoma were similarly short as in ccRCC[11], a consequence of the limited impact of the initiating carcinogens during cancer progression[16,19]. These associations show that mutation rates and their temporal activity are major factors determining phylogenetic tree shapes and sizes.

**Discussion**

With recent success rates of cancer-immunotherapy, understanding the genetic landscapes of immunotherapy-sensitive tumour types and how these influence treatment sensitivity are major needs. dMMR cancers are among the most sensitive solid tumours to checkpoint-inhibiting immunotherapies[5,6] but their genetic evolution, clonal mutation burden and ITH remained unknown. Our series of four treatment-naive dMMR GOAs revealed strikingly high clonal mutation burdens. This may explain the exquisite sensitivity of these cancers to

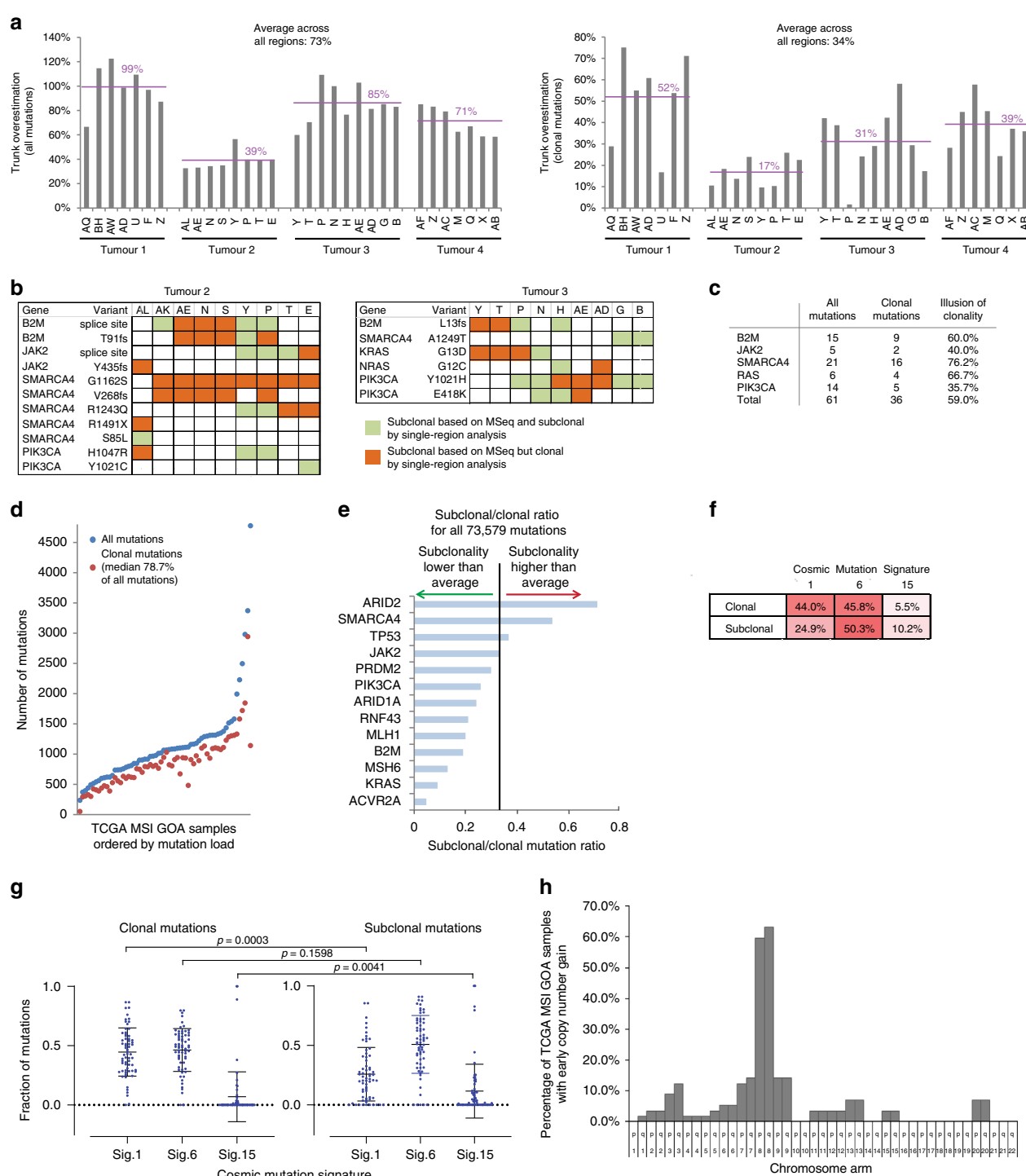

**Fig. 6 Clonal and subclonal mutation analysis. a** Comparison of the total non-silent mutation load per region and of the clonal mutation load per region (defined as present in a cancer cell fraction (CCF) of 0.7 or above) against the number of ubiquitous mutations that have been identified by MSeq. The percent difference to the ubiquitous mutation load is shown. **b** Subclonal driver gene mutations assessed by MSeq in Tumours 2 and 3. In green mutations that single-region analysis picks up as subclonal and in orange mutations that would have been falsely assigned as clonal by single-region analysis. **c** Illusion of clonality (in percent) for driver gene mutations. **d** Mutation load of TCGA MSI GOA samples, number of all mutations in blue and of clonal mutations in red. **e** Subclonal to clonal mutation ratio for driver gene mutations. The black line shows the average ratio across all somatic mutations. **f** Mean percentage of clonal and subclonal mutational signatures found in TCGA MSI GOA samples. **g** Subclonal and clonal mutational signatures for 64 TCGA MSI GOAs. Means and standard deviation are shown and *p*-values have been calculated with a Mann–Whitney test. **h** Percentage of the 64 TCGA samples that gained the indicated chromosome arm early, i.e. before a high number of mutations was acquired through MSI.

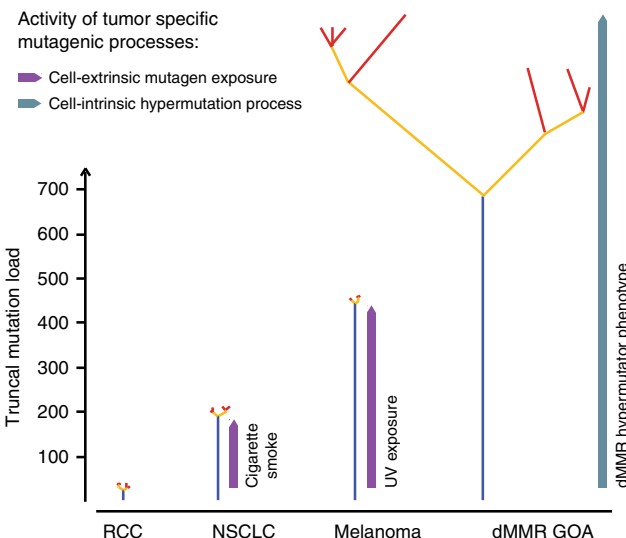

**Fig. 7 Comparison of phylogenetic tree morphologies across four cancer types analysed by MSeq.** Schematics of branched phylogenetic trees drawn with similar branching structures to those directly observed in each of the four tumour types[11,16,19]. Trees were scaled so that trunk and branch lengths are proportional to the average number of ubiquitous and heterogeneous non-silent mutation loads of each tumour type (Fig. 1c). Phylogenetic tree colour code: blue: truncal mutations, yellow: shared mutations, red: private mutations.

immunotherapy as recent data showed that a high clonal mutation burden is a better predictor of immunotherapy success than the total mutation burden[21]. The presence of mutational ITH has furthermore been suggested to impair effective immunotherapy in lung cancer and other malignancies[21,22]. Extremely high numbers of heterogeneous mutations were found in all four dMMR GOAs and these significantly exceeded those in other cancer types analysed by MSeq. Although the analysed tumours were not treated with immunotherapy, these results and the overall high response rate of dMMR GOAs suggests that extreme ITH is unlikely to fundamentally preclude immunotherapy efficacy in tumours with abundant clonal mutations. This warrants MSeq analyses of MSI GOAs that were treated with checkpoint-inhibitors in order to assess whether these hypotheses can be validated in the clinic.

Our study also provides first insights into the clonal origin of lymph node metastatic disease in dMMR GOAs. Lymph nodes were seeded by distinct subclones in the primary tumours, propagating some of the heterogeneity from the primary tumour to metastatic sites. Subclonal mutation generation continued in metastases and similar heterogeneity as observed in primary tumours should therefore be expected in more advanced metastatic disease.

The mutation load of individual tumour regions exceeded the number of truncal mutations by 73%, and still by 34% following subclonal deconvolution. Studies investigating mutation burden as immunotherapy biomarkers may hence benefit from MSeq to robustly and accurately estimate truncal mutation loads. Subclonal immune evasion drivers were identified in two of four cases. Mutations in the *JAK1/2* and inactivation of *B2M* can confer resistance to checkpoint-inhibiting immunotherapy[15,41,42]. Although in MSI colorectal cancer it has been shown that most patients with *B2M* inactivation benefitted from immunotherapy[43], our data suggest that *B2M* loss can be subclonal and is not necessarily propagated to metastases. How subclonal immune evasion drivers and their localization in primary tumours or in metastases impairs immune checkpoint-inhibitor efficacy

in dMMR GOAs should be investigated by MSeq in larger, immunotherapy-treated cohorts.

Despite the selection pressure resulting from the high immunogenicity of dMMR tumours, we found no evidence of reversion of the hypermutator-phenotype. Immune evasion mechanisms which can be readily accessed through single mutations, for example in *HLA* genes, or through biallelic *B2M* or *JAK* mutations may more effectively mitigate against this selection pressure than loss of the dMMR-phenotype, which would still leave behind neoantigen-encoding mutations that have already been generated. Despite considerable mutation loads, cytotoxic T-cell infiltrates were low in two tumours and we could not identify immune evasion events that explain this. This warrants further investigation into immune escape mechanisms in dMMR GOAs.

Defining driver mutations which are commonly truncal is critical for precision cancer medicine approaches as targeting of subclonal driver mutations is likely futile[12]. Several tumour suppressor genes were inactivated by genetic alterations on the trunk in all four tumours. However, loss-of-function of tumour suppressor genes is usually not directly targetable. Two of four dMMR GOAs harboured two inactivating mutations in *ARID1A*. In addition, 25% of MSI GOAs from the TCGA dataset showed two clonal *ARID1A* mutations, further suggesting that biallelic disruption is common. However, given the uncertainty of clonality estimates from single region data, the prevalence of biallelic truncal inactivation will need confirmation by MSeq in larger series. *ARID1A*-deficiency sensitizes cancer cells to small molecule inhibitors of the ATR DNA damage sensor[44]. Such a potential synthetic lethal interaction should be investigated in dMMR GOAs. Additional subclonal mutations in *ARID1A* and in other SWI/SNF-complex members evolved during cancer progression, indicating a role of SWI/SNF-complex modulation during carcinogenesis and cancer progression. MSeq and single sample TCGA data analysis also showed that chromosome 8 gains are among the earliest genetic events in ~60% of these tumours. Further studies are necessary to investigate whether this is relevant for the tolerance of the MSI-phenotype or a marker of aggressiveness as described for other cancer types[45,46].

Comparing results from MSeq analysis and single-region analysis showed that MSeq more accurately identifies clonal and subclonal mutation loads, drivers that are acquired early vs those that evolve late and particularly parallel evolution events. It can furthermore avoid the illusion of clonality of driver mutations and overcome sampling biases which can lead to the failure to accurately identify subclonal driver mutations, for example in *JAK* or *B2M*, that have been suggested to confer therapy resistance[15,41,42]. MSeq should therefore be considered for biomarker discovery in such highly heterogeneous tumour types. Bulk sequencing of DNA from multiple regions and metastases or ctDNA sequencing, followed by bioinformatic identification of clonal mutations are alternative approaches to address the illusion of clonality. MSeq also revealed how the genetic profile of metastatic disease can differ from primary tumours and within different metastatic sites. It finally allowed to assess how selection changes from truncal to private mutations.

Taken together, the dMMR-phenotype remained active throughout the evolution of primary tumours and in metastatic sites, generating extreme ITH. We furthermore revealed the generation of multiple subclonal driver mutations, including remarkable parallel evolution of multiple functionally similar subclonal drivers and a dN/dS ratio indicating positive selection in three of four tumours. These results confirm a high evolvability of dMMR tumours. High heterogeneity and evolvability are thought to enable cancer aggressiveness and poor outcomes[47], yet these data demonstrate a paradoxical association with good prognosis in dMMR tumours. dMMR tumours are unique models to advance insights into cancer

evolution rules and into the potential and current limitations of evolutionary metrics for clinical outcome prediction.

## Methods

**Sample collection and preparation.** Samples from treatment-naive GOA resection specimens were routinely paraffin embedded and fresh frozen at the University Medical Center Hamburg-Eppendorf (Germany). The research use of specimens left over after the pathological diagnosis is regulated through the 'Hamburger Krankenhausgesetz' (Hamburg Hospital Law) in Hamburg, consent and ethical approval are explicitly waived for samples that are fully anonymised. Thus, information about age, sex of the patients and outcome data is not available.

Immunohistochemistry for MLH1, PMS2, MSH2 and MSH6 was performed on 20 cases and four with dMMR (each showing absence of MLH1 and PMS2 staining in cancer cells, see Fig. 1b) were identified by a pathologist. Seven tumour regions representing the spatial extent of each primary tumour were selected (surface area ~8 × 5 mm and a depth of ~10 mm) based on the H&E slide and spatial location within the tumour by a pathologist. Two cases each included two lymph node metastases (Station 1–2, right and left paracardial nodes), which were sufficiently large for analysis. Where necessary, samples were macrodissected to minimize stromal contamination. DNA was extracted using the Qiagen AllPrep kit following the manufacturer's instructions. Nucleic acid yields were determined by Qubit (Invitrogen), and the quality and integrity of DNA was examined by agarose gel electrophoresis. DNA from tumour adjacent non-malignant tissue was used as a source of normal ('germline') DNA. For this, either oesophageal or gastric wall tissue, embedded as "normal mucosa", was chosen and tumour contamination excluded by a pathologist based on H&E slides taken from levels before and after slides for DNA extraction.

**Multiplex immunohistochemistry.** The Opal 7 Tumor Infiltrating Lymphocyte kit (PerkinElmer) was used to perform combined CD8 (antibody dilution 1:300, Opal 570 1:150), pan-Cytokeratin (antibody dilution 1:500, Opal 690 1:150) and DAPI (counter-) stains for each region following the manufacturer's instructions. In Tumour 2, two regions had not enough tissue left after DNA extraction. Slides were scanned using the Vectra 3.0 pathology imaging system (PerkinElmer)[48].

After low-magnification scanning, intratumour regions of interest were scanned at high resolution (20×). Spectral unmixing, tissue and cell segmentation and phenotyping of CD8- and Cytokeratin-positive cells were performed with InForm image analysis software under pathologist supervision. Five representative regions of interests were chosen and cytotoxic T-cells and tumour cells in cancer tissue segmented areas were quantified. From the sum of the five regions, we calculated the ratio of cytotoxic T-cells/tumour cells for each region of Tumours 1–4.

**Whole-exome sequencing.** Tumour and matched germline DNA were sequenced by the NGS-Sequencing facility of the Tumour Profiling Unit at the Institute of Cancer Research. Exome-sequencing libraries were prepared from 1 μg DNA using the Agilent SureSelectXT Human All Exon v6 kit according to the manufacturer's protocol. Paired-end sequencing was performed on the Illumina HiSeq 2500 or NovaSeq 6000 with a minimum target depth of 100× in the adjacent normal samples and a minimum target depth of 200× in tumour regions.

BWA-MEM[49] (v0.7.12) was used to align the paired-end reads to the hg19 human reference genome to generate BAM format files. Picard Tools (http://picard.sourceforge.net) (v2.1.0) MarkDuplicates was run with duplicates removed. BAM files were coordinate sorted and indexed with SAMtools[50] (v0.1.19). BAM files were quality controlled using GATK[51] (v3.5-0) DepthOfCoverage, Picard CollectAlignmentSummaryMetrics (v2.1.0) and fastqc (https://www.bioinformatics.babraham.ac.uk/projects/fastqc/) (v0.11.4).

**Somatic mutation analysis.** Single-nucleotide variant (SNV) calls were generated with MuTect[52] (v1.1.7) and VarScan2[53] (v2.4.1) and mutation calls from both callers were combined. MuTect was run with default settings and post-filtered for a minimum variant frequency of 2%. SNVs generated by MuTect and flagged with 'PASS', 'alt_allele_in_normal' or 'possible_contamination' were retained. SAMtools (v1.3) mpileup was run with minimum mapping quality 1 and minimum base quality 20. The pileup file was inputted to VarScan2 somatic and run with a minimum variant frequency of 2%. The VarScan2 call loci were converted to BED format and bam-readcount (https://github.com/genome/bam-readcount) (v0.7.4) run on these positions with minimum mapping quality 1. The bam-readcount output allowed the VarScan2 calls to be further filtered using the recommended fpfilter.pl accessory script[54] run on default settings. Indel calls were generated using Platypus[55] (v0.8.1) callVariants run on default settings. Calls were filtered based on the following FILTER flags—'GOF, 'badReads, 'hp10', 'MQ', 'strandBias',' Qual-Depth',' REFCALL'. We then filtered for somatic indels with normal genotype to be homozygous, minimum depth ≥10 in the normal, minimum depth ≥20 in the tumour and ≥5 variant reads in the tumour.

The bam-readcount tool was run on all SNV loci using minimum mapping quality 1 and minimum base quality 5 to generate call QC metrics (e.g. average variant base quality, average variant mapping quality). High-confidence SNVs were identified by filtering minimum average variant mapping quality 55 and minimum average variant base quality 35 in called tumour regions based on the

bam-readcount QC metrics. Bam-readcount was then run on the filtered loci using minimum mapping quality 10 and minimum base quality 20 to generate allele counts for the merged VarScan2 and MuTect call loci. All SNV and indel calls were required to have a depth of at least 70 across all tumour regions. SNVs at positions sequenced to less than 20× depth in the matched germline and those which showed a variant frequency in the germline >2% and a variant count >2 were also excluded. Retained mutation calls were then passed through a cross-'germline' filter that flags SNV and indel calls which are present with a VAF of > = 2% in one of fourteen normal samples from the same sample collection. A call is rejected if the variant is flagged as present in 20% or more of the normal samples to remove common alignment artefacts or those arising recurrently at genomic positions which are difficult to sequence. Finally, we applied the following two-tiered filtering strategy to generate MSeq mutation calls. A positive call was made if at least one tumour region had a minimum VAF of 5%. This first tier assures that only mutation calls which have a high probability of being real mutations are selected for further analysis. For any of the mutations that were called in this way, we then determined whether it was present or absent in individual tumour regions. The VAFs for a mutation were looked up with BAM-readcount and a region was called positive if the VAF exceeded 2.5%. Similar two-tier VAF thresholding strategies have been employed in prior MSeq studies[11,16,34]. Private and shared mutations are defined as those that were only detected in a single region or in some but not all tumour regions, respectively, using the minimum VAF of 2.5% as a cutoff. Variant calls on chromosomes X and Y were not considered.

SNV and indel calls were annotated using annovar[56] (v20160201) and oncotator[57] (v1.8.0.0 and oncotator_v1_ds_Jan262015 database) with hg19 build versions. The oncotator 'COSMIC_n_overlapping_mutations' field was used to flag mutations as possible drivers if they occurred in oncogenes and tumour suppressor genes in the online COSMIC Cancer Gene Census (CGC)[58] or in driver genes identified in MSI tumours in the TCGA STAD publication[2]. Mutations were defined as likely driver genes if they led to (1) an amino acid alteration that had previously been described in the COSMIC database, (2) a disrupting mutation, including frameshift-, splice site- or premature stop/nonsense-mutations in a tumour suppressor gene or (3) an amino acid alteration at a position that shows an alteration in the COSMIC CGC but is distinct from the change reported in COSMIC if it was considered a likely driver by the Cancer Genome Interpreter[59].

**DNA copy number aberration analysis.** CNVKit[60] (v0.8.1) was run in non-batch mode for copy number evaluation. Basic target and antitarget files were generated based on the Agilent SureSelectXT Human All Exon v6 kit. Accessible regions suggested by CNVKit (provided in the source distribution as 'access-5kb-mappable.hg19.bed') with a masked HLA interval (chr6:28866528-33775446) form the accessible loci. A pooled normal sample was created from all sequenced germline samples in the series. The copynumber[61] R[62] library functions Winsorize (run with 'return.outliers' = TRUE) and pcf (run with 'gamma' = 200) were used to identify outliers and regions of highly uneven coverage (defined as an absolute log ratio value greater than 0.5) to exclude from the analysis.

We identified high confidence SNP locations using bcftools call[50] (v1.3) with snp137 reference and SnpEff SnpSift[63] (v4.2) to filter heterozygous loci with minimum depth 50. VarScan2 was used to call the tumour sample BAMs at these locations to generate B-Allele Frequency (BAF) data as input for CNVKit. CNVKit was run with matched germline samples along with the adjusted access and antitarget files. For the segmentation step we ran the copynumber function pcf with gamma = 70. Breakpoints were then fed into Sequenza[64] (v2.1.2) to calculate estimates of purity/ploidy and these values were used to recenter and scale the LogR profiles in CNVKit. BAF and LogR profiles were also manually reviewed by two researchers to determine their likely integer copy number states. Adjustments were made in cases where both manual reviews identified a consensus solution that differed from the bioinformatically generated integer copy number profile.

**Cancer cell content, ploidy estimation and wGII.** Cancer cell content was estimated using the scaling factor of the copy number consensus solution. Ploidy was estimated as follows:

$$\text{Ploidy} = (\text{CN}_{\text{Absolute}} \times \text{SegmentLength}) / \sum(\text{SegmentLength}), \qquad (1)$$

with $\text{CN}_{\text{Absolute}}$ representing the unrounded copy number estimate and Segment Length the genomic length between segment break points.

The wGII (ref. [32]) is used to define CIN. We calculated the percentage of integer copy number segments on each chromosome different from the ploidy estimate rounded to the nearest integer state. The percentages are then averaged over the 22 autosomal chromosomes to give the wGII score.

**Subclonality analysis and phylogenetic tree reconstruction.** Allele specific copy number estimates[65] for SNV and indels were calculated as follows:

$$\text{MUT}_{\text{CN}} = \text{VAF}(1/p) \times ((p \times \text{CN}_{\text{Absolute}}) + (2 \times (1-p))), \qquad (2)$$

where VAF is the variant allele frequency and $p$ is the estimated tumour cell content. Cancer cell fraction (CCF) was estimated using the R package Palimpsest[66]. LICHeE[67] was applied to infer phylogenetic trees from the estimated CCF values. The build algorithm was run with CCF/2 as input, -maxVAFAbsent 0,

-minVAFPresent 0.0001 and '-s 10'. In each case, we report the top ranked tree solution. A single valid tree was identified for Tumour 1 (error score: 0.02), Tumour 2 (error score: 0.13) and Tumour 4 (error score: 0.06). LICHeE identified six valid trees for Tumour 3 (error scores: 0.088, 0.095, 0.96, 0.106, 0.112, 0.113). These solutions differed only in the positioning of the branch immediately preceding H2 (which could be positioned at H1 or H3) and of that preceding G2 (which could be positioned at G1) in Fig. 3. The tree with the lowest error score was chosen for the analysis, but selecting any of the alternative solutions would not change any of the conclusions presented in this study. Otherwise, only a low percentage of mutations (2–7% per case) could not be assigned to a subclone in the phylogenetic tree.

Trees were re-drawn and branch lengths scaled to the number of mutations in each subclonal mutation cluster and likely driver mutations were mapped onto the trunk or the appropriate branch. Private mutations identified by LICHeE were split into clonal and subclonal mutations using a CCF threshold of 0.7 unless the algorithm had already identified and split clonal and subclonal clusters. A short branch was added to Tumour 2 following a manual review of the tree solution to represent an 8 mutation cluster that was too small for the algorithm to detect but contained a *B2M* frameshift mutation which was identified as a likely driver.

**Mutational signatures**. All SNV calls were loaded into R using VRanges (v1.28.3)[68] VariantAnnotation, given trinucleotide motifs using SomaticSignatures (v2.18.0)[69] mutationContext and tabulated using motifMatrix with 'normalize' = TRUE. The somatic motifs were then compared with the 30 mutational signatures established in COSMIC[70] V2 using deconstructSigs (v1.8.0)[71] whichSignatures by selecting 'signature.cutoff' = 0 and 'signatures.ref' = 'signatures.cosmic' as run parameters. Mutation signatures representing at least 5% of mutations in one of the analysed mutation groups were reported.

**Ratio of non-synonymous to synonymous mutations (dN/dS)**. We ran dNdScv[40] to generate dN/dS estimates which use trinucleotide context dependent substitution matrices to adjust for common mutation biases. We ran dNdScv with the following optional parameters: 'outp = 1', 'max_muts_per_gene_per_sample = inf' and 'max_coding_muts_per_sample = inf'. This was done separately for mutations shown as truncal (blue), shared (yellow) and private (red and purple) on the phylogenetic trees in Fig. 3.

**HLA mutations and LOH calling**. Mutations in HLA genes were predicted using the program POLYSOLVER[72]. In particular, we first predicted patients' HLA types from germline samples using the shell_call_hla_type script of the POLYSOLVER suite, with the following parameters: race = Unknown, includeFreq = 1 and insertCalc = 0. Then, we used these HLA predictions as input to the shell_-call_hla_mutations_from_type script for predicting HLA mutations in tumour samples. Finally, the shell_annotate_hla_mutations script was used to annotate the mutations identified in the previous step.

LOH events in HLA genes were predicted using the program LOHHLA[39]. LOHHLA requires as input normal HLA types, for which we used POLYSOLVER predictions, along with ploidy and CCF estimates, which were available from the calculations described above. All other parameters were set to default values.

Neopeptides associated to somatic mutations were generated as decribed in ref.[73]. Note that we had to discard ~1.2% of somatic mutations because of inconsistencies between the variant annotation (this can be for either somatic variants or germline variants occurring on the same transcripts as the somatic ones) and the refseq_cds.txt file (GRCh37/hg19 Feb 2009) we used for generating the neopeptides. We used netMHCpan4.0 (28978689) to predict the neopeptides' eluted ligand likelihood percentile rank scores. For each sample, we ran netMHCpan4.0 on all of the samples' neopeptides against all samples' HLA allotypes. As HLA-presented neopeptides, we picked all core peptides (see ref.[73]) with a percentile rank <0.5%.

**MLH1 promoter qPCR**. A total of 250 ng of tumour DNA, CpGenome Human Methylated DNA Standard (Millipore) and CpGenome Human Non-Methylated DNA Standard (Millipore) were subject to bisulphite conversion using the EZ DNA Methylation Gold Kit according to the manufacturer's protocol (Zymo Research Corp.). Methylight primers and probe were used to amplify the *MLH1* CpG island: (forward) 5′-AGGAAGAGCGGATAGCGATTT-3′, (reverse) 5′-TCTTCGTCCC TCCCTAAAACG-3′, (probe) 5′-FAM-CCCGCTACCTAAAAAAATATACGCT TACGCG-BHQ-3′ (ref.[74]). qPCR was performed in a 25-μl reaction with 300 nM primers, 100 nM probe and 1× TaqMan Universal Master Mix II no UNG (Applied Biosystems) using the following program: 50 °C for 2 min, 95 °C for 10 min, followed by 50 cycles at 95 °C for 15 s and 60 °C for 1 min. Samples were analysed in duplicate in 96-well plates on an AB QuantStudio 6 Flex RT-PCR System.

**Mutation loads and clonal/subclonal drivers in TCGA MSI GOAs**. Sixty-four GOAs from TCGA cohort[2] are classified as MSI in the cBIO web portal[75]. We downloaded the BAM files of these cases from the NIH GDC Legacy Archive[76]. Adjustments to the analysis steps were necessary due to the properties of the TCGA sequencing data. A minimum variant frequency of 5% was applied throughout the mutation calling and the fpfilter.pl parameters 'min-ref-avrl' and 'min-var-avrl'

filters were relaxed to 50. The minimum depth requirement in the tumour sample was relaxed to 20, while the minimum average base and mapping quality were set to 20 and 40, respectively. No adjustments were made to the default access and antitarget files of the CNVkit analysis due to large variations in the sequencing depths of the normal samples across the cohort. Otherwise, the somatic mutation, copy number and subclonality analysis steps were as described above. Mutational signatures were run as before and those detected with a mean contribution of 5% or more further analysed.

**Reporting Summary**. Further information on research design is available in the Nature Research Reporting Summary linked to this Article.

## Data availability

The multi-region exome-sequencing data have been deposited in the European Genome-Phenome archive under the accession code EGAS00001003434. The TCGA gastroesophageal dataset referenced during the study is available from the NIH GDC Data Portal website (https://portal.gdc.cancer.gov). All the other data supporting the findings of this study are available within the Article and its Supplementary Information files and from the corresponding author upon reasonable request. A Reporting Summary for this Article is available as a Supplementary Information file.

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

## Acknowledgements

The study was supported by a Wellcome Trust Strategic Grant (105104/Z/14/Z) to the ICR Centre for Evolution and Cancer, by the National Institute for Health Research Biomedical Research Centre for Cancer at the ICR/RMH, by a Clinician Scientist Fellowship from Cancer Research UK and by grants from Cancer Genetics UK and the Constance Travis Trust.

## Author contributions

K.v.L. processed the tissue, designed and conducted experiment, analysed the data and wrote the paper. A.W. performed bioinformatics analyses, analysed the data and wrote the paper. M.P. and S.L. performed bioinformatics analyses. B.G. processed the tissue. L.B., M.S., G.Sp. and B.C. analysed the data. K.F. and N.M. conducted exome sequencing. R.S., A.M. and G.Sa. provided the tissue and performed dMMR analysis. M.G. designed the study, supervised the experiments and data analysis and wrote the paper. All authors read and approved the manuscript.

## Competing interests

The authors declare no competing interests.
