## [Peer Review File · Nature Communications]

Reviewers' comments:

Reviewer #1 (Remarks to the Author):

Review Extreme intratumour heterogeneity and driver evolution in mismatch repair
2 deficient gastro-oesophageal cancer

The authors improved the manuscript substantially, mainly:

1. They corrected the dN/dS analysis.
2. They expanded to comparison with TCGA GOAs MSI cases.
3. They added the analysis of clonal vs sub-clonal mutation and their relationships to private vs truncal mutations.
4. They discussed single region vs MSeq sequencing.
5. They added neoantigens predictions analysis.

Overall, I support publishing this manuscript in N. Communication. Mainly because of its finding of the convergence evolution in different clones and positive selection (dN/dS ratio >1) in the private mutations.

I still think however that they should flip the order of the comparison in the section called "Comparison of multi-region vs single-regions heterogeneity analysis" and instead of showing how TCGA data supports their finding emphasize what are the new finding the TCGA cannot support. For example L348-L340 " Together with a significant increase in signature 15 among subclonal mutations, this supports the change in mutational processes between early progression and subclonal diversification as seen in the MSeq dataset."

1. L67-L68 "Microsatellite instable (MSI) and DNA mismatch repair deficient (dMMR) cancers are a distinct subtype of GOAs". Are not MSI and DNA mismatch repair deficient cancers the same thing?
2. L94-L95 ACVR2A MSH3 and PRDM2 are also highly mutated e.g. Maruvka et al NBT 2017 DOI: 10.1038/nbt.3966
3. L103-L104. Cannot the 'illusion of clonality' be solved by bulk sequencing of the entire tumor together as a bulk? Or the different regions?
4. L164-L166 "No other mutational signatures contributed substantially to the heterogeneous mutations" You show that tumor 3 has about 10% of Sig 14 that was shown recently (Haravdala et al N. Comm 2017 DOI: 10.1038/s41467-018-04002-4) to be due to POLE exonuclease domain mutation in addition to dMMR. This can explain the higher mutation burden tumor 3 has.
5. L191 "have a copy number which is lower" did you mean allele fraction ?
6. L226-L228 Both PRDM2 and ACVR2A were suggested as drivers in GOAs MSI by Maruvka et al NBT 2017.
7. Can you hypothesize why only tumors 1&2 shows a dN/dS>?
8. Fig5, The shared mutations in tumor 4 shows a deviation from 1 in the dN/dS to a similar extent as 1&2 private mutations. Can you elaborate on that? Do you think it is a sign of negative selection?
9. L318. I'm still a little confused. Is every thing that you gain from MSeq is dealing with the 'illusion of clonality' ? Am I wrong or everything else can more or less be detected via bulk sequencing? What about the convergence evolution? This is something that is very hard or impossible to be inferred from a single region sequencing
Also see point 3 above.
10. L390-L394. I personally find this to be a major finding that cannot or almost cannot be inferred from single region sequencing even if this is a very large region.

Reviewer #2 (Remarks to the Author):

I thank the authors for their efforts to address my original comments. The manuscript has improved considerably and I do not longer have any major concerns. The size of the cohort is limited, as I originally noted, but the analyses and the findings are interesting.

Minor comments:

Lines 136-137. Please review this sentence.

Lines 192-193. Consider editing this sentence to improve clarity. Perhaps changing "showed a near complete absence of mutations" to "showed a near complete absence of mutations in two copies".

Line 316. I suggest removing "strongly" from the sentence considering that the 95% confidence intervals that do not overlap 1 are close to it and that multiple confidence intervals are shown without multiple testing adjustment (e.g. Benjamini, 2005; JSTOR 27590520).

Reviewer #3 (Remarks to the Author):

Most of the comments from the previous submission have been address and the language surrounding the importance of the work is more appropriate.

Some issues remain:

- Lines 123-127: Is the comparison with other cancer correct conceptually? The authors study MSI cancer that by their nature are highly mutated. A comparison with "normal" tumours of kidneys and lung should yield statistical difference (as would a comparison of other gastric cancers with MSI cancers). It is rather self-fulfilling.
- Lines 223-224: Potential hypermethylation of MLH1 promoter can be readily tested by qPCR to test this hypothesis
- Line 248: The whole "Identification of parallel evolution" section is poorly cited and very speculative. For example, the mutation in PIK3CA is postulated "that the same mutation independently evolved twice, once in AL and once in the ancestor cell of P1 and Y1". It highly unlikely that exactly (to a nucleotide) the same mutation arose in different regions of the tumour. I think it is more likely that AL, being a metastatic sample, was seeded from an ancestor of samples P and Y.
- Lines 386-389: The link between immunotherapy and results presented in the manuscript is overstated. The experiments presented here do not support or reject the hypothesis.
- Lines 490-520: This methods section states 2% cut-off for VAF frequency with the exception of the final statement using 5% cut-off. However, it is not clear to me how this is used. "Finally, calls were retained if at least one tumour region had a variant frequency of at least 5% and the same call was reported in all other regions of the same tumour showing a variant frequency of at least 2.5%." In my opinion this indicates that no mutations should be called private as it is essential for the mutation to have 2.5% VAF in all samples. Also, in the response to comment 5 of reviewer 2, 5% cut-off is mentioned but is described as in methods.
- Figure 6D (page 31): Figure legends says 21.3% of all mutations. What number does it refer to? It seems that red dots on the plot are equal to around 70-90% of the blue dots.

Potential typo? Text on line 344 suggests that subclonal (not clonal) mutations are equal to 21.3%.

Point by point reply to the reviewer's comments

We would like to thank the reviewers for their thorough and supportive comments. We have addressed all comments and provide a detailed reply that also indicates the changes that we made in the manuscript.

Reviewer #1:

The authors improved the manuscript substantially, mainly:

1. They corrected the dN/dS analysis.
2. They expanded to comparison with TCGA GOAs MSI cases.
3. They added the analysis of clonal vs sub-clonal mutation and their relationships to private vs truncal mutations.
4. They discussed single region vs MSeq sequencing.
5. They added neoantigens predictions analysis.

Overall, I support publishing this manuscript in N. Communication. Mainly because of its finding of the convergence evolution in different clones and positive selection (dN/dS ratio >1) in the private mutations.

Reviewer comment: I still think however that they should flip the order of the comparison in the section called "Comparison of multi-region vs single-regions heterogeneity analysis" and instead of showing how TCGA data supports their finding emphasize what are the new finding the TCGA cannot support. For example L348-L340 " Together with a significant increase in signature 15 among subclonal mutations, this supports the change in mutational processes between early progression and subclonal diversification as seen in the MSeq dataset."

Author reply:

We have restructured this paragraph which is now putting more emphasis on the new findings that were only possible through MSeq and we changed the order of the panels in Fig. 6 accordingly:

"Having established these limitations of single-region heterogeneity analyses, we analysed the 64 MSI GOAs cancers from TCGA. All samples harboured subclonal mutations. However, only a median of 21.3% of mutations were subclonal (Fig. 6D) compared to a median of 60.1% in the MSeq data. We then assessed the clonality of mutations in driver genes which we had found to be either predominantly clonal or subclonal by MSeq. The highest frequency of subclonal mutations was found in *ARID2* and *SMARCA4* whereas *ACVR2A* was almost always clonal in TCGA data (Fig. 6E), consistent with MSeq data where these occurred late and early, respectively. Mutations in the remaining driver genes were predominantly clonal in TCGA data, but in light of our MSeq data this is likely limited by the overestimation of clonal status in single-region analysis.

Only 2/64 (3.1%) TCGA cases showed parallel evolution of two subclonal *SMARCA4* mutations, each, and two subclonal *PIK3CA* mutations evolved in one case. No parallel evolution of driver mutations in *RAS* or immune evasion regulators was

identified. Together with the detection of parallel evolution in spatially distinct tumour regions by MSeq, this illustrates the limitation to identify such events by single sample analysis. Of note, two independent disrupting mutations in *ARID1A* were found to be clonal in each of 16 out of 64 tumours (25%) and only 4 tumours had one clonal and one subclonal inactivating event. Combined with our observation of truncal *ARID1A* alterations, this may support early biallelic inactivation in a fraction of these tumours.

Mutational signature analysis showed that clonal and subclonal mutations were dominated by MSI-specific mutational signatures 6 and 15 (Fig. 6F-G), confirming our MSeq results. 44.0% of clonal mutations displayed signature 1 and although this significantly decreased among subclonal mutations, it remained the second most abundant mutation signature. Together with a significant increase in signature 15 among subclonal mutations, this supports the change in mutational processes between early progression and subclonal diversification as seen in the MSeq dataset. Timing of copy number changes in the TCGA dataset furthermore supported that chromosome 8 gains had been acquired before or early after the MSI-phenotype in ~60% of cases (Fig. 6H and Supplementary Fig. 10)."

Reviewer comment: 1. L67-L68 "Microsatellite instable (MSI) and DNA mismatch repair deficient (dMMR) cancers are a distinct subtype of GOAs". Are not MSI and DNA mismatch repair deficient cancers the same thing?

Author reply:

The terms microsatellite instable (MSI) and mismatch repair deficient (dMMR) cancers are indeed often used interchangeably. More formally, MSI is predominantly used when microsatellite instability (or multiple of its associated features such as high mutation loads and indel rates) has been identified by genetic techniques whereas dMMR is mainly used to describe tumours where IHC showed loss of mismatch repair proteins. In line with these conventions we used the term dMMR to describe our cases which were identified by IHC and MSI for those studies that had defined tumours as MSI based on mutation and indel rates etc.

Reviewer comment: 2. L94-L95 ACVR2A MSH3 and PRDM2 are also highly mutated e.g. Maruvka et al NBT 2017 DOI: 10.1038/nbt.3966

Author reply:

Thank you for pointing out this reference. We added it to the text.

Reviewer comment: 3. L103-L104. Cannot the 'illusion of clonality' be solved by bulk sequencing of the entire tumor together as a bulk? Or the different regions?

Author reply:

We agree that bulk sequencing of the entire cancer could be an alternative to multi-region sequencing for the detection of clonal and subclonal aberrations. ctDNA sequencing may also be able to achieve this. We added this to the discussion:

“Combining DNA from multiple tumour regions and metastatic sites and to then apply bulk sequencing followed by bioinformatic identification of clonal mutations or the use of ctDNA sequencing are alternative approaches to address the illusion of clonality, sampling biases and to estimate clonal mutation burdens more accurately.”

Reviewer comment: 4. L164-L166 “No other mutational signatures contributed substantially to the heterogeneous mutations”. You show that tumor 3 has about 10% of Sig 14 that was shown recently (Haravdala et al. N. Comm 2017 DOI: 10.1038/s41467-018-04002-4) to be due to POLE exonuclease domain mutation in addition to dMMR. This can explain the higher mutation burden tumor 3 has.

Author reply:

Thank you for pointing out this publication. Tumour 3 harboured indeed a non-synonymous POLD1 (P107T) mutation but this was only present in regions B, G, H, N1, P2, AD, AE and T2 but not in the regions P1, T1, Y and N2. The mutation was hence subclonal whereas signature 14 was only detected in clonal but not in subclonal mutations. The POLD1 mutation can therefore not explain the clonal signature 14 mutations and the absence of subclonal signature 14 mutations in regions harbouring the POLD1 mutation points against a functional role of the P107T amino acid change. We have added the above reference and the following explanation to the text:

“10.5% of the ubiquitous mutations in Tumour 3 showed signature 14 which has been described in mismatch repair deficient cancers that are also *POLE* or *POLD1* mutant²⁹. Tumour 3 harboured a *POLD1* mutation (P107T) which has not previously been described, but this was subclonal and could hence not explain the presence of clonal signature 14 mutations. The absence of signature 14 from subclonal mutations furthermore suggests that POLD1 P107T may be a passenger mutation. “

Reviewer comment: 5. L191 “have a copy number which is lower” did you mean allele fraction?

Author reply:

This refers to the ‘mutation copy number’ which has been defined by McGranahan, Sci Transl Med, 2015 (DOI: 10.1126/scitranslmed.aaa1408) as the number of alleles harbouring the mutation. We have now used the term ‘mutation copy number’ in the text to clarify this and added the reference to directly link to detailed information on this method.

“Copy number gains that occurred early can be identified if the majority of mutations in that region have a mutation copy number²⁴ which is lower than that of the gained allele.”

Reviewer comment: 6. L226-L228 Both PRDM2 and ACVR2A were suggested as drivers in GOAs MSI by Maruvka et al NBT 2017.

Author reply:

Thank you for pointing this publication out. We have now added this to the text:

“Both genes have been suggested as likely drivers in MSI GOAs¹⁴.”

Reviewer comment: 7. Can you hypothesize why only tumors 1&2 show a dN/dS>?

Author reply:

We believe that the number of cases is too small to definitively assess this. However, the identification of parallel evolution – another signal of positive selection – in Tumour 3 which does not show an increase in dN/dS suggests that patterns of selection either differ between cancers or that dN/dS is not sensitive enough to detect selection in some. We added this to the text:

“The partial overlap of these two indicators of positive selection suggests that patterns of selection differ between cancers or that the sensitivity of dN/dS analysis may be too low to detect positive selection in some cases.”

Reviewer comment: 8. Fig5, The shared mutations in tumor 4 shows a deviation from 1 in the dN/dS to a similar extent as 1&2 private mutations. Can you elaborate on that? Do you think it is a sign of negative selection?

Author reply:

This could indicate negative selection and we now describe this further:

“The decrease in dN/dS to 0.83 in the shared mutations of Tumour 4 may be a sign of negative selection during early cancer evolution. Our results show that MSeq allows to dissect the temporal dynamics of selection in dMMR tumours and this can be used to reveal what genetic alterations are selected for or against in larger series.”

Reviewer comment: 9. L318. I’m still a little confused. Is every thing that you gain from MSeq is dealing with the ‘illusion of clonality’? Am I wrong or everything else can more or less be detected via bulk sequencing? What about the convergence evolution? This is something that is very hard or impossible to be inferred from a single region sequencing. Also see point 3 above.

Reviewer comment: 10. L390-L394. I personally find this to be a major finding that cannot or almost cannot be inferred from single region sequencing even if this is a very large region.

Author reply to points 9 and 10:

We made the advantages from MSeq clearer with an additional paragraph in the discussion:

“Comparing results from MSeq analysis and single-region analysis showed that MSeq more accurately identifies clonal and subclonal mutation loads, drivers that are acquired early vs those that evolve late and particularly parallel evolution events. It can furthermore avoid the illusion of clonality of driver mutations and overcome

sampling biases which can lead to the failure to accurately identify subclonal driver mutations, for example in *JAK* or *B2M*, that have been suggested to confer therapy resistance^{16,43,44}. MSeq should therefore be considered for biomarker discovery in such highly heterogeneous tumour types. Combining DNA from multiple tumour regions and metastatic sites and to then apply bulk sequencing followed by bioinformatic identification of clonal mutations or the use of ctDNA sequencing are alternative approaches to address the illusion of clonality, sampling biases and to estimate clonal mutation burden. MSeq also revealed how the genetic profile of metastatic disease can differ from primary tumours and within different metastatic sites. It finally allowed to assess how selection changes from truncal to private mutations.”

Reviewer #2:

I thank the authors for their efforts to address my original comments. The manuscript has improved considerably and I do not longer have any major concerns. The size of the cohort is limited, as I originally noted, but the analyses and the findings are interesting.

Minor comments:

Reviewer comment: 1. Lines 136-137. Please review this sentence.

Author reply:

Thank you for pointing out that this was confusing. We have changed it to:

“A median of 1194 mutations were only detectable in some but not in all analysed tumour regions per case and hence heterogeneous.”

Reviewer comment: 2. Lines 192-193. Consider editing this sentence to improve clarity. Perhaps changing "showed a near complete absence of mutations" to "showed a near complete absence of mutations in two copies".

Author reply:

We appreciate this clarification and have changed the sentence to the following wording to address this:

“...showed a near complete absence of mutations on all copies of the gained allele...”

Reviewer comment: 3. Line 316. I suggest removing "strongly" from the sentence considering that the 95% confidence intervals that do not overlap 1 are close to it and that multiple confidence intervals are shown without multiple testing adjustment (e.g. Benjamini, 2005; JSTOR 27590520).

Author reply:

We have changed the text accordingly.

Reviewer #3:

Most of the comments from the previous submission have been address and the language surrounding the importance of the work is more appropriate.

Some issues remain:

Reviewer comment: 1. Lines 123-127: Is the comparison with other cancer correct conceptually? The authors study MSI cancer that by their nature are highly mutated. A comparison with “normal” tumours of kidneys and lung should yield statistical difference (as would a comparison of other gastric cancers with MSI cancers). It is rather self-fulfilling.

Author reply:

Assessing how MSI influences tumour evolution in comparison to tumours without known hypermutator mechanisms or with temporally restricted mutagen exposure was one of our main aims. We indeed expected MSI tumours to be more heterogeneous and to have higher mutation loads than some of the others. Our results confirmed this and provide the first quantitative analysis of the differences in clonal and subclonal mutation loads in these tumours. We believe that this is an important addition to prior analyses that compared mutation loads of tumour types with different mutational mechanisms based on single sample sequencing (e.g. Alexandrov, Nature, 2013, (DOI: 10.1038/nature12477)).

Reviewer comment: 2. Lines 223-224: Potential hypermethylation of MLH1 promoter can be readily tested by qPCR to test this hypothesis

Author reply:

We have now applied *MLH1* promoter methylation testing and this confirmed methylation in Tumours 2-4. This is now mentioned in the results section and the qPCR data has been added as Supplementary Figure:

“Hypermethylation of the *MLH1* promoter as the cause for dMMR in these cases was confirmed with a qPCR methylation assay (Supplementary Fig. 6)³².”

Details on the method were added to the methods section:

MLH1 promoter qPCR

250 ng tumour DNA, CpGenome Human Methylated DNA Standard (Millipore) and CpGenome Human Non-Methylated DNA Standard (Millipore) were subject to bisulfite conversion using the EZ DNA Methylation Gold Kit according to the manufacturer’s protocol (Zymo Research Corp.). Methylight primers and probe were used to amplify the *MLH1* CpG island: (forward) 5'-AGGAAGAGCGGATAGCGATTT-3', (reverse) 5'-TCTTCGTCCTCCCTAAAACG-3', (probe) 5'-FAM-CCCCTACCTAAAAAATATACGCTTACGCG-BHQ-3'⁷⁶. qPCR was performed in a 25 µl reaction with 300 nM primers, 100 nM probe and 1x TaqMan Universal Master Mix II no UNG (Applied Biosystems) using the following program: 50 C for 2 min, 95 C for 10 min, followed by 50 cycles at 95 C for 15 s and 60 C for 1 min. Samples were analysed in duplicate in 96-well plates on an AB QuantStudio 6 Flex RT-PCR System.”

Reviewer comment: 3. Line 248: The whole “Identification of parallel evolution” section is poorly cited and very speculative. For example, the mutation in PIK3CA is postulated “that the same mutation independently evolved twice, once in AL and once in the ancestor cell of P1 and Y1”. It highly unlikely that exactly (to a nucleotide) the same mutation arose in different regions of the tumour. I think it is more likely that AL, being a metastatic sample, was seeded from an ancestor of samples P and Y.

Author reply:

These conclusions were drawn based on the phylogenetic analysis which used accepted phylogenetic tree generation algorithms and standard phylogenetic interpretation as applied by us and others in the past (for example Gerlinger, NEJM, 2012 (DOI: 10.1056/NEJMoa1113205) and Nature Genetics, 2014 (DOI: 10.1038/ng.2891), de Bruin, Science, 2014 (DOI: 10.1126/science.1253462), Jamal-Hanjani, NEJM, 2017 (DOI: 10.1056/NEJMoa1616288), Turajlic, Cell, 2018 (DOI: 10.1016/j.cell.2018.03.043), Yates Cancer Cell, 2017 (DOI: 10.1016/j.ccell.2017.07.005). We have added some of these as additional references to the text to support our approach.

More specifically, the conclusions that two identical PIK3CA mutations evolved in parallel in Tumour 2 were reached based on the following data:

The phylogenetic tree for Tumour 2 shows that the metastatic site AL diverged early from all other samples that were analysed and that the last common ancestor of AL and P1/Y1 is also the common ancestor of all other tumour regions. Based on the phylogenetic tree, any mutation that is shared by AL and P1/Y1 should therefore

A) either also be shared by all other tumour regions if it was acquired by a single event in this last common ancestor

or

B) have been acquired independently by AL and the clade that included P1 and Y1

As the PIK3CA mutation is not shared by all tumour regions and clones, scenario B is the likely explanation of the two mutations according to the phylogenetic tree.

Another possibility is that the mutation was acquired on the trunk and subsequently lost by some subclones. This is a hypothesis that is not explicitly tested by phylogenetic algorithms. For this reason, all versions of the manuscript already stated that we did not find evidence of copy number losses that would support this.

That seeding of AL from P1 or Y1 is unlikely can furthermore be concluded when looking at the mutation numbers that were detected in these subclones: AL harbours the clonal PIK3CA mutation and 237 clonal mutations that were not found in any other tumour region (represented by the length of the red branch section of AL). The last common ancestor of all other tumour regions (i.e. excluding AL) had acquired 69 mutations (common orange branch). P1 and Y1/3, S, N, AE and AK furthermore acquired 126 mutations (common orange branch) from their last common ancestor.

Thus, if region AL (containing a clonal PIK3CA mutation) was seeded from the PIK3CA mutant clone in regions P1/Y1, it would have had to physically lose these 195 clonal

mutations as none of them is detectable in AL. Back-mutation of 195 mutations to wild type status within one clone appears more unlikely than the independent acquisition of the same mutation twice in this tumour. This logic is incorporated in the mathematics behind phylogenetic tree reconstruction and we therefore did not include these considerations into the text.

However, we agree that an explanation as to why the same mutation can evolve twice would be helpful and added the following sentence:

“The acquisition of the same mutation twice may appear unlikely, but a tumour of this diameter is likely to contain >10 billion cancer cells¹⁰ that have undergone approximately the same number of cell divisions to grow to this size from the founding cell. It is conceivable that two cells independently acquire the same mutation in some tumours of this size.”

Reviewer comment: 4. Lines 386-389: The link between immunotherapy and results presented in the manuscript is overstated. The experiments presented here do not support or reject the hypothesis.

Author reply:

We agree that no clinical implications for immunotherapy allocation can be drawn from our findings in tumours that were not treated with checkpoint inhibitors. However, we present a novel hypothesis for further testing and we made this clear by adding the following sentence to the discussion:

“This warrants MSeq analyses of MSI GOAs that were treated with checkpoint inhibitors in order to assess whether this hypothesis can be validated in the clinic.”

Reviewer comment: 5. Lines 490-520: This methods section states 2% cut-off for VAF frequency with the exception of the final statement using 5% cut-off. However, it is not clear to me how this is used. “Finally, calls were retained if at least one tumour region had a variant frequency of at least 5% and the same call was reported in all other regions of the same tumour showing a variant frequency of at least 2.5%.” In my opinion this indicates that no mutations should be called private as it is essential for the mutation to have 2.5% VAF in all samples. Also, in the response to comment 5 of reviewer 2, 5% cut-off is mentioned but is described as in methods.

Author reply:

We thank the reviewer for highlighting that our description of the variant calling approach was not clear.

VarScan2 and MuTect are routinely run with a standard 2% VAF threshold for initial mutation calls in our pipeline as we process samples with various tumour contents and sequencing depths. We then apply the following two-tiered approach for MSeq data analysis:

A mutation call needs to have a minimum VAF of 5% in at least one region to be accepted as a positive call. This first tier assures that only mutation calls which have a high probability of being real mutations are selected for further analysis and avoids the inclusion of false positives. The cutoff of 5% was chosen based on the analyses by Cibulskis, *Nature Biotechnology*, 2013 (DOI: 10.1038/nbt.2514) which showed that the sensitivity of MuTect drops rapidly for variants below 5% VAF at 200x coverage.

For any of the mutations that were called in this way, we then determine whether it is present or absent in individual tumour regions. The VAFs for each mutation call are looked up with BAM-readcount and a region is called positive if the VAF exceeds 2.5% and negative if it is below 2.5%. This step is important as standard mutation calling often misses mutations with VAF <5% and therefore often overestimates heterogeneity for variants that just cross the 5% VAF threshold in one or few samples. A minimum VAF threshold is necessary in this step to avoid overcalling at the very low VAF spectrum where false positives increase due to the intrinsic error rates of NGS. Unfortunately, there is no reliable technique to interpret these very low VAF variants in NGS data unless error correction approaches, such as barcodes or duplex sequencing, are used.

This two-tier calling approach has been used by us and others in prior MSeq studies (for example de Bruin, *Science*, 2014 (DOI: 10.1126/science.1253462), Jamal-Hanjani, *NEJM*, 2017 (DOI: 10.1056/NEJMoa1616288), Gerlinger, *Nature Genetics*, 2014 (DOI: 10.1038/ng.2891)). de Bruin used a minimum VAF of 5% in a single tumour region and a minimum of 2 variant reads in any region for a presence call (equivalent to 2% VAF at the 100X sequencing depth). In the current dMMR MSeq study, we used a 2.5% threshold instead of a fixed minimum number of variant reads to avoid that the VAF detection limit changes between tumours with different sequencing depths.

We hope that this clarifies the calling setup and we have updated the methods section to avoid misunderstanding:

“Finally, we applied the following two-tiered filtering strategy to generate MSeq mutation calls. A positive call was made if at least one tumour region had a minimum VAF of 5%. This first tier assures that only mutation calls which have a high probability of being real mutations are selected for further analysis. For any of the mutations that were called in this way, we then determined whether it was present or absent in individual tumour regions. The VAFs for a mutation were looked up with BAM-readcount and a region was called positive if the VAF exceeded 2.5%. Similar two-tier VAF thresholding strategies have been employed in prior MSeq studies^{12,17,36}.”

Please note that a private mutation is defined as a mutation only called in a single region with a VAF of at least 5% but absent in all other regions (i.e. not exceeding 2.5% VAF when looked up by BAM-readcount in other regions). We have added the following to the methods section to clarify that this is limited by the sensitivity of the sequencing and calling approach:

“Private and shared mutations are defined as those that were only detected in a single region or in some but not all tumour regions, respectively, using the minimum VAF of 2.5% as a cutoff.”

Reviewer comment: 6. Figure 6D (page 31): Figure legends says 21.3% of all mutations. What number does it refer to? It seems that red dots on the plot are equal to around 70-90% of the blue dots.

and

Potential typo? Text on line 344 suggests that subclonal (not clonal) mutations are equal to 21.3%.

Author reply:

We apologize that the figure incorrectly mentioned 21.3% of clonal mutations. This is actually the number of subclonal mutations. We have revised Figure 6 and added the correct median percentage for clonal mutations.

REVIEWERS' COMMENTS:

Reviewer #1 (Remarks to the Author):

The authors made a good job addressing my and the other reviewers comments. I think that in addition to the new results presented here, Mseq described here can be applied in larger and other cohorts could lead to a meaningful increase in our understanding of tumor evolution. Congratulations!

Reviewer #2 (Remarks to the Author):

The authors have incorporated the minor changes that I requested and I have no additional comments.

Reviewer #3 (Remarks to the Author):

The authors have addressed all the comments raised from our review.